# Characteristics of the Mg-Zn-Ca-Gd Alloy after Mechanical Alloying

**DOI:** 10.3390/ma14010226

**Published:** 2021-01-05

**Authors:** Sabina Lesz, Bartłomiej Hrapkowicz, Małgorzata Karolus, Klaudiusz Gołombek

**Affiliations:** 1Department of Engineering Materials and Biomaterials, Silesian University of Technology, 44-100 Gliwice, Poland; bartlomiej.hrapkowicz@polsl.pl (B.H.); klaudiusz.golombek@polsl.pl (K.G.); 2Institute of Materials Engineering, University of Silesia, 40-007 Katowice, Poland; karolus@us.edu.pl

**Keywords:** metallic alloys manufacturing, Mg-based alloy, mechanical alloying, SEM, XRD

## Abstract

Magnesium-based materials are interesting alternatives for medical implants, as they have promising mechanical and biological properties. Thanks to them, it is possible to create biodegradable materials for medical application, which would reduce both costs and time of treatment. Magnesium as the sole material, however, it is not enough to support this function. It is important to determine proper alloying elements and methods. A viable method for creating such alloys is mechanical alloying, which can be used to design the structure and properties for proper roles. Mechanical alloying is highly influenced by the milling time of the alloy, as the time of the process affects many properties of the milled powders. X-ray diffraction (XRD) and scanning electron microscopy (SEM) with energy-dispersive spectroscopy (EDS) were carried out to study the powder morphology and chemical composition of Mg_65_Zn_30_Ca_4_Gd_1_ powders. Moreover, the powder size was assessed by granulometric method and the Vickers hardness test was used for microhardness testing. The samples were milled for 6 min, 13, 20, 30, 40, and 70 h. The hardness correlated with the particle size of the samples. After 30 h of milling time, the average value of hardness was equal to 168 HV and it was lower after 13 (333 HV), 20 (273 HV), 40 (329 HV), and 70 (314 HV) h. The powder particles average size increased after 13 (31 μm) h of milling time, up to 30 (45–49 μm) hours, and then sharply decreased after 40 (28 μm) and 70 (12 μm) h.

## 1. Introduction

The development of biomedical materials has significantly contributed to the progress of medicine and implantology [1,2]. Each new material continues to affect patient treatment, reducing treatment costs and time. Therefore, the requirements for materials are growing, and they must meet certain criteria to allow their use. These requirements are all the more important the more complex the implant is. Currently, the largest research is being conducted on biodegradable materials [3,4].

Magnesium-based materials have promising mechanical properties and potential that they can serve as implants, hence, they are widely researched to serve as such. Since their mechanical properties are similar to those of a human bone, this makes them a perfect solution for biodegradable fracture fixtures [5]. This is particularly important as the bone growth, and therefore the regrowth as well, is at least partially connected with the mechanical properties of the osseous tissue [6,7,8,9]. It is hypothesized to be caused by microscopic damage stimulating bone adaptation, as explained by Wolff’s law, i.e., if higher mechanical load on a certain bone is present, then, the bone will adapt and remodel itself to be stronger in order to resist said load accordingly. Moreover the inverse is also true, i.e., while the load on a bone decreases, the density and the strength of that bone will also decrease, which may lead to a condition known as osteopenia [10,11,12]. Thus, in order to maintain similar loads on the implant and on the bone, the material properties should be as similar to bone as possible, an important criterion fulfilled by light magnesium alloys. Magnesium is characterized by low density (1.79 g/cm^3^). It is the fourth most abundant cation in the whole human body, respectively of its great physiological importance. In the body, 50% of Mg is situated in bones, 49% inside the tissues and organs cells, and barely 1% in the blood. It has been claimed that Mg influences the nervous system, muscle work, and the reinforcement of bone structure while it supports processes occurring in the bones, such as the regulation of the relaxation and muscle cramps. The possibility of using pure magnesium, without any impurities, seems to be attractive for medical purposes. The magnesium alloys are prized for their combination of lightness and strength, high corrosion resistance, and good biocompatibility [5]. Good mechanical properties of these alloys are achieved through chemical homogeneity of the synthesized alloys and admissible content of impurities. Mechanical alloying is a process, which ensure the previously mentioned advantage of magnesium-based alloys. Although it is known that pure magnesium has poor mechanical strength, as an alloy it can be vastly improved and possess more attractive characteristics such as good biocompatibility and biodegradability [13]. Due to those traits, an implant made of magnesium alloys would increase a patient wellbeing, reduce the healing time, and reduce cost due to making the re-operation for removing the temporary implant obsolete [14]. However, magnesium alloys are usually not recommended for such applications due to the toxicity of the other elements. Hence, the alloy composition should be taken into consideration, for both reasons, such as the strength and the toxicity. However, the possible choices of alloying elements are gravely limited, due to biocompability problems [15]. Magnesium reacts extremely well with zinc, which increases its corrosion resistance and mechanical properties [16,17], moreover, it is relatively harmless to the human body [18]. Another element worth mentioning is calcium as it affects the microstructure, mechanical properties, electrochemical behavior, and kinetics of the degradation of magnesium-based implants [19]. In addition trace quantities of low toxicity rare earth elements, such as gadolinium can be tolerated by the human organism, which additionally affect the mechanical strength of the alloy [20,21].

Staiger et al. reported the formation of Mg(OH)_2_ oxide layer on crystalline magnesium based-alloys, however, in the presence of chloride ions this layer was quickly destroyed promoting rapid degradation [22]. In addition, bubbles of hydrogen resulted from the magnesium corrosion, which could provoke pain, discomfort, and even death in the worst-case scenario [23]. It was found that over 20% Zn addition could reduce the hydrogen evolution to acceptable levels [24]. However, in crystalline magnesium alloys it is not possible to obtain such solubility of zinc due to the formation of different intermetallic phases, but it is possible in bulk metallic glasses (BMG’s). Bulk metallic glasses are characterized by an amorphous structure and can be obtained by solidification or solid-state processing. It has been reported that amorphous materials have superior corrosion resistance to their crystalline counterparts [25]. Henceforth, amorphous magnesium alloys are better in terms of zinc solubility, effectively decreasing the H_2_ evolution rate, and in corrosion resistance as well.

In spite of the abovementioned facts, it is reported that, in the current state, the amorphous magnesium alloys, also named as magnesium bulk metallic glasses (BMG’s), are known to be lacking the structural, and therefore mechanical stability. Although very promising, such disadvantages take them out of the picture for the orthopedic and dentistry applications [26]. Moreover it is difficult to manufacture such alloys due to vast sensibility of the glass-forming ability to the preparation processes and eventual impurities, which may result in drastic deterioration of said trait [15]. Amorphous alloys are usually developed by solidification and because the process is faster and easier, it may be manufactured by solid-state processing. One such method is mechanical alloying (MA), which involves obtaining fine structures in processed alloys [27,28]. It has been reported that both nanocrystalline and amorphous alloys can be obtained [29,30,31]. Moreover it allows for the homogeneous alloy formation caused by the material transfer in the MA process [32]. Mechanical alloying involves a blend of powders and/or particles subjected to highly energetic compressive impact forces in a high-energy mill. Then, the alloys are formed by repeated processes of cold welding and fracturing of the powder particles. High-energy mechanical alloying (HEMA) is a relatively simple method of producing alloy powder, which can be consolidated using various manufacturing techniques, thanks to advances in laser treatment such as, for example, selective laser sintering, selective laser melting, etc. [33,34]. Moreover, it renders complicated preparations processes obsolete, such as complicated melting procedures [31,35]. Therefore, the advantage of this method justifies this method as a green alternative to the conventional foundry method. Therefore, this approach is attracting interest as a result of environmental and sustainability advantages. Furthermore, the ability to obtain different alloys in a powdered form allows the abovementioned techniques of consolidation to be used to obtain different shapes and textures of a detail, such as an implant. This allows for the creation of a specific tool tailored to meet all the needs of a patient, increasing the odds of a successful treatment, as well as decreasing costs and time as previously mentioned. As reported by Salleh et al., the milling time of the alloy changes its properties considerably, yet such conclusions are not widely reported [14,36]. Hence, the objective of this paper is to present a preliminary study on the effect of high-energy milling times on the quaternary Mg_65_Zn_30_Ca_4_Gd_1_ alloy. X-ray diffraction and scanning electron microscopy with energy-dispersive spectroscopy were carried out to study powder morphology and chemical composition of Mg_65_Zn_30_Ca_4_Gd_1_ powders. The powder size was assessed by the granulometric method and the Vickers hardness test was used for microhardness testing.

## 2. Materials and Methods

Alloy powders with a nominal composition of Mg_65_Zn_30_Ca_4_Gd_1_ were processed by mechanical alloying with a mixture of pure powders Mg (99.99% wt.%), Zn (99.99% wt.%), Gd (99.99% wt.%), and Ca pieces (99.99% wt.%). All materials were purchased from Alfa Aesar (Heysham, UK). The base materials were weighed and enclosed in a stainless steel container under high-purity argon (99.99% wt.%) atmosphere. The ball-to-powder ratio was set to 10:1. The SPEX 8000D (Metuchen, NJ, USA) high energy shaker ball mill was used for MA at room temperature. The shaking mode was used with a 30 min stop every 1 h. The samples were prepared by alloying in a constant frequency shaker ball milling for varying milling times, i.e., 6 min., 13, 20, 30, 40 and 70 h, respectively. The process is illustrated in the schematic in Figure 1.

The X-ray diffraction (XRD) analyses were performed with the use of the Empyrean PANalytical Diffractometer (Almelo, The Netherlands) equipped with a copper anode (Kα1 λCu = 1.5418 Å), Ni filter, and PIXcell detector. The measurements were made in a step-scanning mode in a range of 10–150° 2θ. The phase analysis of substrates and milling products were performed with a High Score Plus PANalytical software (Almelo, The Netherlands) integrated with the ICDD PDF4+ 2016 data base (International Centre for Diffraction Data, Newtown Square, PA, USA). Zeiss 35 scanning electron microscope (SEM) (Carl Zeiss, Jena, Germany) with a voltage of 20 kV, equipped with energy-dispersive spectroscopy (EDS) was used to determine the morphology of the obtained powders after 6 min. and 13, 20, 30, 40, and 70 h of milling time. Particle size distribution of powders was measured using the Fritsch Analyssette 22 MicroTec + (Weimar, Germany) in ethyl alcohol. The hardness test was performed on Future-Tech FM700 Vickers hardness tester (Kawasaki, Japan) with 15 s dwell time and 50 g of force (HV0.05). Every sample was measured with 5 indentations. The size of the impression was measured with the aid of a calibrated microscope with a tolerance of ±1/1000 mm [37].

## 3. Results and Discussion

### 3.1. Phase Composition

Milling processes are known to considerably alter the morphology of milled powders as a result of severe plastic deformation. During mechanical alloying, the same process occurs, repeatedly fracturing and welding the powder particles. Short milling times were found to have no pronounced effect on both the particles and overall alloy composition. In various studies, it has been reported that the milling time could influence the alloy phase composition, resulting in an amorphous phase [31,38]. Hence, the milling time was chosen based on the reports of Mg_65_Zn_30_Ca_5_ alloy amorphization [31,38]. However, in these mentioned experiments another diameter of milling balls was used, whereas while using 10 mm balls the amorphous structure was verified after 13, 20, 30, 40, and 70 h for Mg_65_Zn_30_Ca_5_ alloy. Therefore, samples of the Mg_65_ZnCa_4_Gd_1_ were milled for 6 min., 13, 20, 30, 40, and 70 h of milling time in order to assess the differences. The X-ray diffraction patterns, presented in Figure 2, show a presence of an amorphous halo. The broad scattering maxima between 35° and 45° 2 theta confirm an amorphization process, which is attributed to the diffusive mixing mechanism [31]. The amorphization for the magnesium based-alloys was confirmed by Datta [31,38] and Lesz [31,38] in their respective works. After 6 min, the XRD pattern shows the presence of pure Mg, Zn, Ca(OH)_2_, and GdH_2_ with the beginning stage of MgZn_2_ secondary phase formation. The crystalline phases observed on the X-ray diffraction patterns correspond to MgZn_2_, GdH_2_, Mg, MgO, Zn, and Ca(OH)_2_ phases. The MgO and Ca(OH)_2_ phases appeared due to the exposure of the samples to oxygen in the air between tests. After 13 h, extending the milling time, there were no visible changes in the form of diffraction patterns which remained almost the same.

The Mg phase may be present due to the solid-state diffusion being in an incomplete stage, whereas gadolinium hydride had appeared due to the short air exposition between analyses. Amorphization has been observed on a large number of alloy systems below a critical crystallite size of a few nanometers having been synthesized by high-energy ball-milling technique [31,32,39].

### 3.2. Granulometry and Hardness

Figure 3 represents cumulative volume curve distributions (curve Figure 3a–e) and the general volume distribution (histograms Figure 3a–e) showing the particle size volume share for powder, milled for 13, 20, 30, 40, and 70 h, respectively. All samples maintain the range of particles from 10 to 100 μm with irregular, non-equiaxial shapes. The median value (Q_3_(x), where x = 50%) (Q3(x) 50%) of the particle sizes are 31 μm for 13 h, and 45, 49, 28, and 12 μm for 20, 30, 40, and 70 h, respectively. It can be clearly seen, that after 13 and 40 h of milling the particle size is similar, however, it sharply decreases after 70 h. For better visualization of the results Q_3_(x), where x = 10% and 90% are shown on their respective graphs.

The average particle size values with error bars are presented in Table 1 and the hardness values (with error bars) are presented below in Table 2. All of the results were presented as the mean ± standard deviation (SD).

In the case of the described alloys (Table 1), when using the SPEX 8000 high-energy shaker ball mill for MA at room temperature, we expect that the initial grain growth (20 and 30 h) is associated with the initial stage of the alloying process, i.e., the formation of intermetallic phases. With the increase in grinding time (40 and 70 h), the grain size reduction process, which is typical for MA. Similar effects have been observed in similar works [32,39,40].

Average values of hardness measurements, as well as particle size, for Mg_65_Zn_30_Ca_4_Gd_1_ powder alloy milled for 13, 20, 30, 40, and 70 h are presented in Figure 4. The average values of hardness are 333 ± 24, 273 ± 25, 168 ± 22, 329 ± 29, and 320 ± 11 HV after 13, 20, 30, 40, and 70 h, respectively. The results are presented in Table 2. Correlating the values of the samples’ hardness with their average particle size, it can be said that after a certain amount of time the powders particles are fragmented more thoroughly, thus, decreasing in size. During this process, strong plastic deformation occurs due to constant friction welding and fragmentation which results in an increase in the hardness value [32,39,40,41,42,43]. Clearly, after 30 h of milling time, the hardness is lower that after 20 h, but it can be correlated to the particle size, as the size after 30 h is bigger as well [32,39,41].

Mechanical alloying can be usually divided into a few stages [41,42,43]. In the first stage of the mechanical alloying process the particle sizes are both smaller and bigger than the initial particles. There is wide hardness dispersion because some of the powder particles are trapped between the grinding balls and show strong deformation while the others are intact.

The second stage is the merging of the particles, which results in larger particles. Usually, the hardness increases. This can be seen in Figure 4, which represents the average hardness values as well as particle size for Mg_65_Zn_30_Ca_4_Gd_1_ powder alloy milled for 13, 20, 30, 40, and 70 h, respectively

The third stage is when random welding orientations occur, leading to a rolled structure composed of equiaxial particles, therefore, decreasing the plasticity.

In the last stage, the narrow dispersion of particles size occurs leading to the saturation of hardness. This stages can and will occur cyclically. This can be seen when comparing the results of both particle size and hardness test for 13 and 70 h. Obviously, after 70 h, the particles are smaller than after 13, albeit the hardness is comparable.

### 3.3. Scanning Electron Microscopy: Morphology and Qualitative Chemical Composition Analysis

Scanning electron microscopy analysis revealed similarities in the morphology of all powders, as it can be seen in Figure 5. On the SEM micrographs, there are visible irregular shapes with some agglomerations. Those assemblies of finer particles may be attributed to the number of hours during mechanical alloying, as it consists of repetitive cold welding and grinding of said powders. It can be already visible after 13 h (Figure 5a) on the larger particles. Moreover, it is worth noting that the changes occurring during the mechanical alloying are of a statistical nature. Clearly, it is caused by unique deformation processes which every ground particles are affected by [41,44]. After 20 h, the particles consolidate, thus, seemingly they increase in size, although they get more uniform in the process. Whereas after 13 h, the particles may be smaller, yet clearly consist of heterogeneous “chunks”. This phenomenon occurs due to deformed particles joining together, hence, the particle size distribution curve is pushed toward the right side of the graph, indicating larger sized particles. However, after 30 h of milling, the particles sharply decrease, which can be seen in Table 1. The particle sizes visible on SEM micrographs are reflected by the performed granulometry.

EDS results from the micrographs featuring the measured area for the Mg_65_Zn_30_Ca_4_Gd_1_ powder alloy samples after 13, 20, 30, 40, and 70 h (Figure 5) are presented in Table 3.

As it can be seen from the atomic weight distribution, after 13 h, the chemical composition stabilizes, meaning the solid-state diffusion process reaches its peak. Hence, from 13 h to 70 h the chemical composition is almost identical (statistically negligible) to the assumed composition of Mg_65_Zn_30_Ca_4_Gd_1_. Henceforth, it can be stated that, after 13 h, the particles are chemically homogeneous, which is an important factor for medical applications [40]. The results of the EDS analysis (Table 3) show a good correlation with the assumed composition of the Mg_65_Zn_30_Ca_4_Gd_1_ alloy. Whereas the initial stage of the mechanical synthesis (13 h of grinding) indicates the presence of unreacted components (Figure 2), the results obtained for the alloy after the maximum grinding time (70 h), indicating an overestimated amount of magnesium, refer to the material in which the share of the amorphous phase increases, in which the interface border disappears, and the distribution of elements may have local heterogeneities. The time needed for homogeneous redistribution of the alloying elements in the magnesium matrix during alloy synthesis depends on diffusion mobility of alloying elements, and also on the type of alloying elements, their synergistic influence, and phase composition. Due to significant deformations in the milled particles, the recovery process associates with the dislocation movement. As a result of the recovery, the mechanical properties and the structure of the metal observed in the microscopic scale do not change substantially.

## 4. Conclusions

The morphology of the milled powders reveals that the particles vary in the range from 12 to 50 μm average values, corresponding to the granulometric tests. The powder particles average size increases after 13 h of milling time, equaling 31 μm and up to 30 h equaling 45–49 μm. Then, sharply decreases after 40 and 70 h of milling time and equaling 28 μm and 12 μm, respectively. The particle size has a strong influence on the hardness of the samples. After 30 h of milling time, the average values of hardness is equal to 168 HV, which is lower than the hardness of the rest samples that are about 300 HV. During the milling process, the particles are subjected to repeated welding and fracturing. Strong plastic deformations occur, thus, increasing the hardness, while at the same time the particle size decreases. The fluctuations in the particle size (and therefore hardness) are characteristic for the MA process, as they are repeatedly subjected to cold welding, fracturing and grinding. This cyclic process leads to homogeneity in chemical composition, which is supported by the EDS. After 13 h, the chemical composition stabilizes, and the phase composition is not change by further milling time. After 13 and 70 h, there is no significant change in hardness, meaning that at those stages the hardness satiation for that alloy was reached.

Therefore, it can be concluded that MA is a utility for achieving homogeneity of the alloy and microstructure refinement, which is desirable for enhancing the alloy’s mechanical performance and corrosion behavior.

## Figures and Tables

**Figure 1 materials-14-00226-f001:**
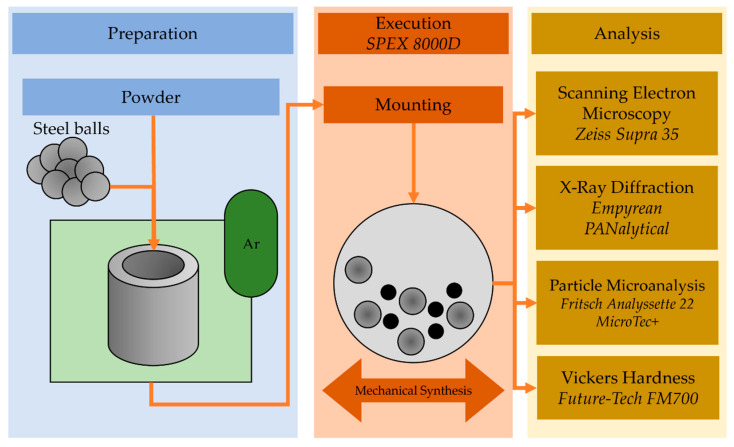
Schematical representation of the experimental process.

**Figure 2 materials-14-00226-f002:**
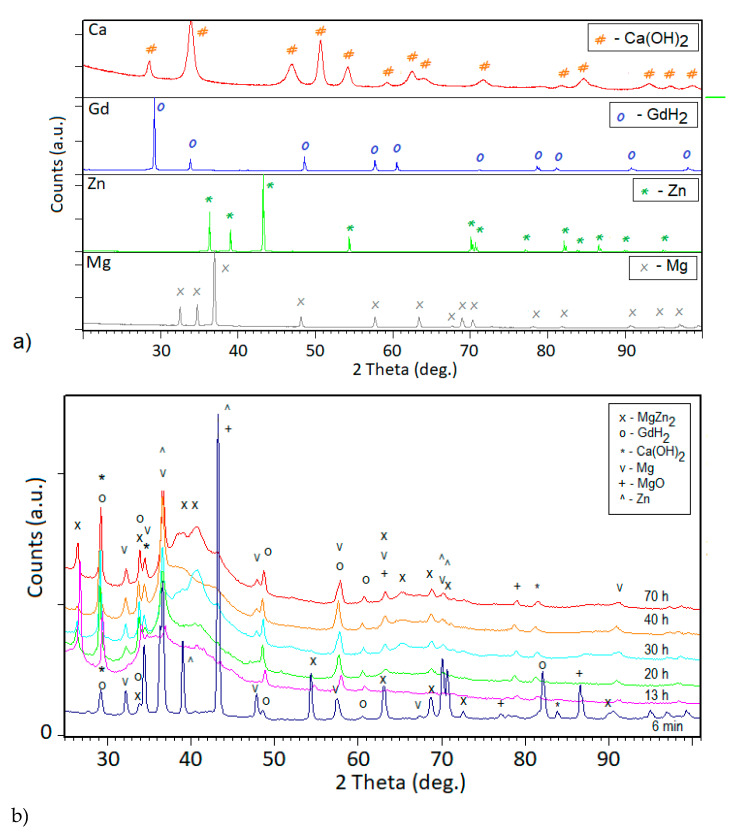
The X-ray diffraction patterns obtained for (**a**) Ca, Gd, Zn, Mg initial powders; (**b**) Mg_65_Zn_30_Ca_4_Gd_1_ powder alloy samples synthesized after 6 min, 13, 20, 30, 40, and 70 h of milling time, respectively. The most important peaks of main identified phases are labeled according to the ICDD PDF4+ 2016 data base (MgZn_2_, 03-065-0120; GdH_2_, 01-089-4063; Ca(OH)_2_, 00-004-0733; Mg, 00-001-1141; MgO, 00-003-0998; Zn, 03-065-3358).

**Figure 3 materials-14-00226-f003:**
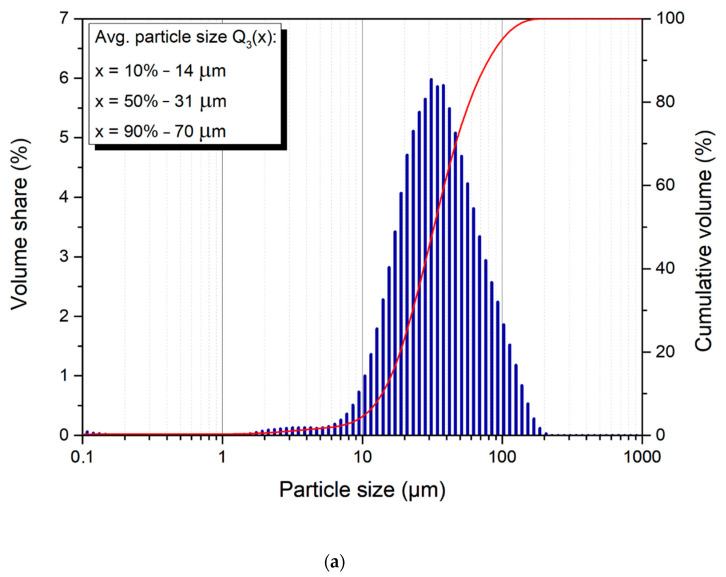
Particle size volume share (histogram) and their cumulative distribution (curve) for Mg_65_Zn_30_Ca_4_Gd_1_ powder alloy samples synthesized for (**a**) 13; (**b**) 20; (**c**) 30; (**d**) 40; (**e**) 70 h, respectively.

**Figure 4 materials-14-00226-f004:**
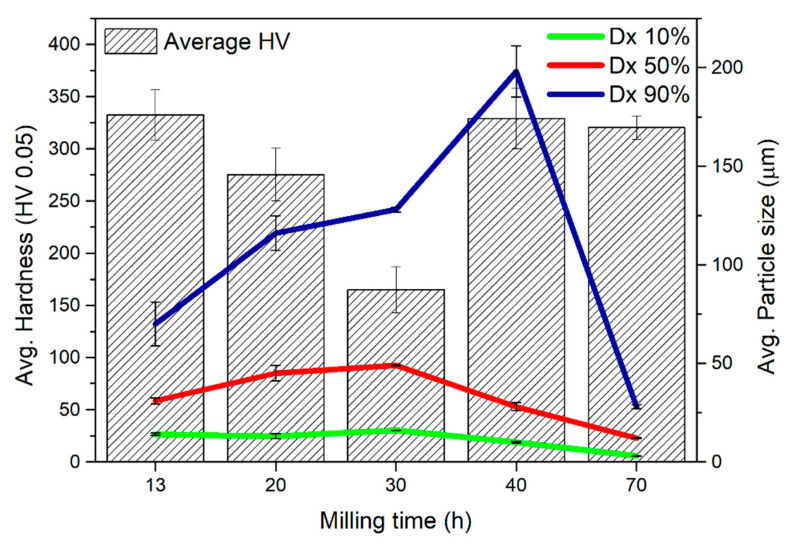
Graph representing average hardness values, as well as particle size, for Mg_65_Zn_30_Ca_4_Gd_1_ powder alloy samples milled for 13, 20, 30, 40, and 70 h, respectively.

**Figure 5 materials-14-00226-f005:**
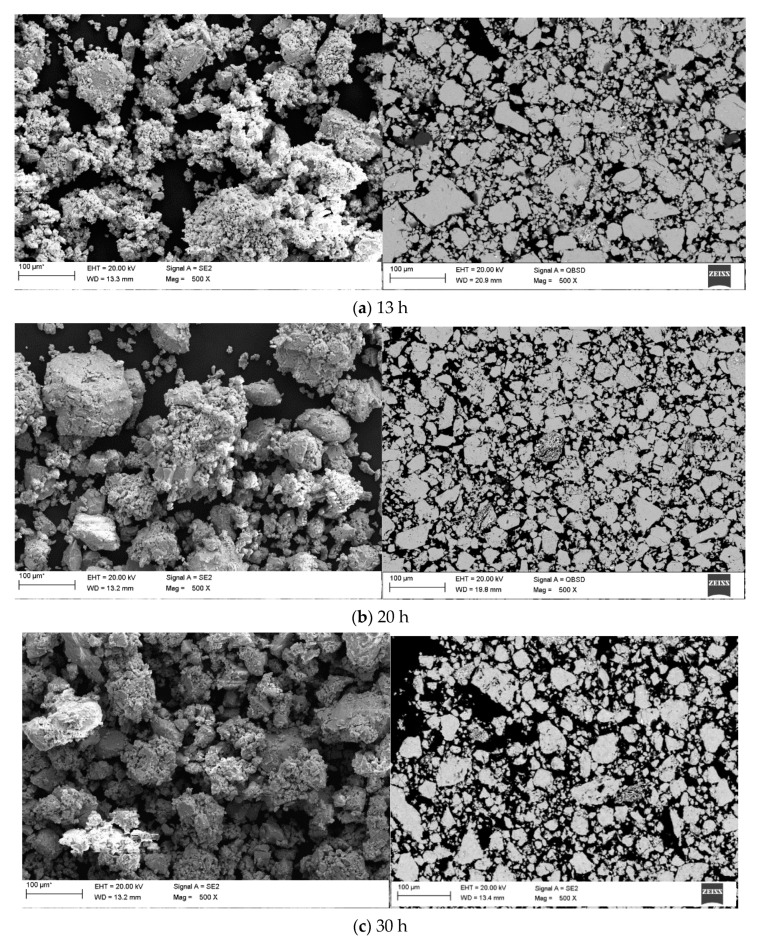
SEM (scanning electron microscopy) results-Morphology with the micrographs of the Mg_65_Zn_30_Ca_4_Gd_1_ alloy powder samples milled for (**a**) 13; (**b**) 20; (**c**) 30; (**d**) 40; (**e**) 70 h, respectively.

**Table 1 materials-14-00226-t001:** Average particle size values with error bars for samples obtained for 13, 20, 30, 40, and 70 h of milled time.

Sample	Average Particle Size (μm)
13 h	31 ± 1.5
20 h	45 ± 4.0
30 h	49 ± 0.3
40 h	28 ± 2.1
70 h	12 ± 0.3

**Table 2 materials-14-00226-t002:** Hardness value with error bars for samples obtained for 13, 20, 30, 40, and 70 h of milled time.

Test Number	13 h Sample (HV 0.05)	20 h Sample (HV 0.05)	30 h Sample (HV 0.05)	40 h Sample (HV 0.05)	70 h Sample (HV 0.05)
1	305	273	179	359	333
2	305	253	141	348	316
3	343	249	151	348	316
4	367	282	153	285	303
5	343	319	201	305	333
Average:	333 ± 24	275 ± 25	168 ± 22	329 ± 29	320 ± 11

**Table 3 materials-14-00226-t003:** EDS (energy-dispersive spectroscopy) results from the Mg_65_Zn_30_Ca_4_Gd_1_ powder alloy samples for 13, 20, 30, 40, and 70 h, respectively.

Sample (Milling Time)	(wt.%)				(at.%)			
	Mg	Ca	Gd	Zn	Mg	Ca	Gd	Zn
13	39 ± 1.9	12 ± 0.6	4 ± 0.2	45 ± 2.2	61 ± 3.1	11 ± 0.6	1	27 ± 1.4
20	40 ± 2.0	6 ± 0.3	5 ± 0.3	50 ± 2.5	64 ± 3.2	6 ± 0.3	1	29 ± 1.5
30	40 ± 2.0	3 ± 0.1	4 ± 0.2	52 ± 2.6	65 ± 3.3	3 ± 0.1	1	31 ± 1.6
40	40 ± 2.0	3 ± 0.1	4 ± 0.2	52 ± 2.6	65 ± 3.3	3 ± 0.1	1	31 ± 1.6
70	43 ± 2.10	4 ± 0.2	4 ± 0.2	49 ± 2.5	67 ± 3.4	4 ± 0.1	1	28 ± 1.4
Theoretical value	40.94	4.15	4.08	50.83	65.00	4.00	1.00	30.00

## Data Availability

Data sharing is not applicable to this article.

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
