# Peer review of "Characteristics of the Mg-Zn-Ca-Gd Alloy after Mechanical Alloying"

_materials, 2021, doi:10.3390/ma14010226_

Round 1
Reviewer 1 Report
Magnesium-based alloy systems are widely studied these days, Mg-Zn-Ca-Gd alloy is promising candidate for bio applications. This review paper focuses on mechanical alloying method and its characterization. The topic itself is of great importance, the manuscript is well organized and comprehensively described. Below are major issues that the reviewer would encourage the authors to consider in further improving this manuscript.
1, Please consider changing the title from “Characteristic of the Mg-Zn-Ca-Gd alloy after mechanical alloying” to “Characteristics of the Mg-Zn-Ca-Gd alloy after mechanical alloying”.
2, Please provide more information about your synthesis process (2. Materials and Methods).Please also add more details to your Figure 1, include names of instruments you have used, e.g., SPEX 8000D mixer mill.
3, As you have mentioned, the hardness value is related with plastic deformation and all related with particle sizes, please add more background information and citations of other groups for line 187-line196. Also add more details of your hardness measurements set up.
4, Please consider revise your manuscript, the English of the paper is not yet at a professional level , e.g., “In order to study the structure of the Mg65Zn30Ca4Gd1 powders methods such as X-ray diffraction for the phase analysis, scanning electron microscopy with energy-dispersive spectroscopy in order to determine the powder morphology and chemical composition were used. ” Should be changed to “X-ray diffraction and scanning electron microscopy with energy-dispersive spectroscopy were carried out to study powder morphology and chemical composition of Mg65Zn30Ca4Gd1 powders.”
Author Response
Dear Reviewer,
Thank you very much for your suggestions. All of them have been taken into consideration in order to improve this paper.
Please note that the referees’ comments are given in italic and my replies are given in bold. The changes in the text of the revised manuscript are highlighted in colour.
All my responses to your comments are highlighted using the "Track Changes" function in Microsoft Word.
- Please consider changing the title from “Characteristic of the Mg-Zn-Ca-Gd alloy after mechanical alloying” to “Characteristics of the Mg-Zn-Ca-Gd alloy after mechanical alloying”.
Our reply: We thank you for this comment. According to the Reviewer’s comments, the title was changed.
Old title: Characteristic of the Mg-Zn-Ca-Gd alloy after mechanical alloying
New title: Characteristics of the Mg-Zn-Ca-Gd alloy after mechanical alloying
- Please provide more information about your synthesis process (2. Materials and Methods). Please also add more details to your Figure 1, include names of instruments you have used, e.g., SPEX 8000D mixer mill.
Our reply: We thank you for this comment. According to the Reviewer’s comments, more information was provided. Line 121, page 3
The ball-to-powder ratio was set to 10:1. The SPEX 8000D high energy shaker ball mill was used for MA at room temperature. The shaking mode with a 30 min stop every 1 hour was used.
Fig. 1. was changed. The names of devices were added.
- As you have mentioned, the hardness value is related with plastic deformation and all related with particle sizes, please add more background information and citations of other groups for line 187-line196. Also add more details of your hardness measurements set up.
Our reply: We thank you for this comment. According to the Reviewer’s comments, an explanation was added. Statement in lines 209-211 (page 9).
During this process strong plastic deformation occurs due to constant friction welding and fragmentation which results in an increase of the hardness value [32,39–43]. Clearly after 30 hours of milling time the hardness is lower that after 20 hours, but it can be correlated to the particle size, as the size after 30 hours is bigger as well.
Mechanical alloying can be usually divided in few stages [41–43].
More details of hardness measurements were added: from line 141, page 5:
The hardness test was performed on Future-Tech FM700 Vickers hardness tester with 15 seconds dwell time and 50 grams of force (HV0.05). Every sample was measured with 5 indentations. The size of the impression is measured with the aid of a calibrated microscope with a tolerance of ±1/1000 mm [37].
- Please consider revise your manuscript, the English of the paper is not yet at a professional level , e.g., “In order to study the structure of the Mg65Zn30Ca4Gd1 powders methods such as X-ray diffraction for the phase analysis, scanning electron microscopy with energy-dispersive spectroscopy in order to determine the powder morphology and chemical composition were used. ” Should be changed to “X-ray diffraction and scanning electron microscopy with energy-dispersive spectroscopy were carried out to study powder morphology and chemical composition of Mg65Zn30Ca4Gd1 powders.”
Our reply: Thank you very much for your suggestions. The English has been substantially improved (the manuscript has been approved by an English-speaking native). All my responses to your comments are highlighted using the "Track Changes" function in Microsoft Word.
For example:
In abstract:
“In order to study the structure of the Mg65Zn30Ca4Gd1 powders methods such as X-ray diffraction for the phase analysis, scanning electron microscopy with energy-dispersive spectroscopy in order to determine the powder morphology and chemical composition were used”.
Was changed to:
The X-ray diffraction and scanning electron microscopy with energy-dispersive spectroscopy were carried out to study powder morphology and chemical composition of Mg65Zn30Ca4Gd1 powders.”
In Materials and Methods:
“In order to determine the morphology of the obtained powders after 6 min. and 13, 20, 30, 40 and 70 hours of milling time were observed using the Zeiss 35 scanning electron microscope equipped with energy-dispersive spectroscopy.”
Was changed to:
“Zeiss 35 scanning electron microscope equipped with energy-dispersive spectroscopy was used to determine the morphology of the obtained powders after 6 min. and 13, 20, 30, 40 and 70 hours of milling time”.
Thank you again for all your suggestions and comments. I hope you appreciate the specific changes I have made in response to these comments and that overall you feel that the main arguments and contribution are now much stronger.
Sincerely yours,
Sabina Lesz

Reviewer 2 Report
In their manuscript the authors investigate the preparation of high alloyed Mg-Zn-Ca-Gd alloys suitable for biodegradable applications in medical environments. Alloy powders are prepared by mechanical alloying, and particle size, global chemical composition and hardness are characterized as a function of alloying time.
The paper is thoroughly written. However, prior to publication some further questions should be addressed:
1) In the comprehensive introduction the authors state that biodegradable materials in e.g. implants should resemble “properties as close to the bone as possible”. Since bones are not built of Mg alloys it should be stated which specific properties the authors target, here.
2) The authors observe a sequence of decreasing, increasing and further decreasing grain size of the alloys, changing with alloying time. This is ascribed to cyclic welding and fracturing of chemically homogeneous particles during the alloying process. However, I would assume that these processes occur statistically in the mixture of particles, and not subsequently in a defined time order to all particles at once. Then, however, I would expect homogeneous and stable particle size distributions after some onset time, if only welding and fracturing are involved. From my point of view, alternative processes such as dynamic recrystallization and inhomogeneous element distributions (segregations) on the local scale, that change with milling time, might be responsible for the observed sequence. Can the authors comment on this? What is the local element distribution in the overall chemically homogeneous particles?
3) In Tables 1 and 2 as well as in Fig. 4 error bars for the average alloy properties should be added.
Author Response
Dear Reviewers,
Thank you very much for your suggestions. All of them have been taken into consideration in order to improve this paper. All my responses to your comments are highlighted using the "Track Changes" function in Microsoft Word.
- In the comprehensive introduction the authors state that biodegradable materials in e.g. implants should resemble “properties as close to the bone as possible”. Since bones are not built of Mg alloys it should be stated which specific properties the authors target, here.
Our reply: We thank for this comment. According to the Reviewer’s comments, an explanation was added. Statement in lines 47-57 (page 2).
Magnesium is characterized by low density (1.79 g/cm3). It is the fourth most abundant cation in the whole human body, respectively of its great physiological importance. In the body, 50% of Mg is situated in bones, 49% inside the tissues and organs cells, and barely 1% in the blood. It is claimed, that Mg has influence on the nervous system, muscle work and the reinforcement of bone structure while it supports processes occurring in the bones, such as the regulation of the relaxation and muscle cramps. The possibility of using pure magnesium, without any impurities, seems to be attractive for medical purposes. The magnesium alloys are prized for its combination of lightness and strength, high corrosion resistance, good biocompatibility [38]. Good mechanical properties of these alloys are achieved through chemical homogeneity of the synthesized alloys and admissible content of impurities. Mechanical alloying is process, which can ensure mentioned earlier advantage of magnesium based alloys.
- The authors observe a sequence of decreasing, increasing and further decreasing grain size of the alloys, changing with alloying time. This is ascribed to cyclic welding and fracturing of chemically homogeneous particles during the alloying process. However, I would assume that these processes occur statistically in the mixture of particles, and not subsequently in a defined time order to all particles at once. Then, however, I would expect homogeneous and stable particle size distributions after some onset time, if only welding and fracturing are involved. From my point of view, alternative processes such as dynamic recrystallization and inhomogeneous element distributions (segregations) on the local scale, that change with milling time, might be responsible for the observed sequence. Can the authors comment on this? What is the local element distribution in the overall chemically homogeneous particles.
Our reply: We strongly agree with the Reviewer's opinion that the observed cyclic sequence of increasing and decreasing the grain size of the alloys is related to many aspects. The processes taking place in the material during mechanical alloying generate various effects such as crushing, melting, and dynamic recrystallization. Undoubtedly, the inhomogeneous distribution may result in local changes in the properties of the material obtained.
However, in the case of the described alloys (Table 1), when using the SPEX 8000 high energy shaker ball mill for MA at room temperature, we expect that the initial grain growth (20, 30 h) is associated with the initial stage of the alloying process, i.e. the formation of intermetallic phases. With the increase in grinding time (40, 70 h), the grain size reduction process, which is typical for MA. Similar effects are observed in similar works [32,39,40]. (Statement in lines 177- 181, page 8).
In the case of the studied alloys, we determined the chemical composition of selected grains using point analysis. In order to ensure statistically correct results of the chemical composition and distribution of elements in the material, EDS analysis were performed for 6 points and 4 areas.
The results of the EDS analysis (Table 3) show a good correlation with the assumed composition of the Mg65Zn30Ca4Gd1 alloy. Whereas the initial stage of the mechanical synthesis (13 h of grinding) indicates the presence of unreacted components (Figure 2). While, the results obtained for the alloy after the maximum grinding time (70h), indicating an overestimated amount of magnesium, refer to the material in which the share of the amorphous phase increases, in which the interface border disappears, and the distribution of elements may have local heterogeneities. The time needed for homogeneous redistribution of the alloying elements in the magnesium matrix during alloy synthesis depends not only on diffusion mobility of alloying elements, but also on the type of alloying elements, their synergistic influence, and phase-composition. Due to significant deformations in the milled particles, the recovery process associates with the dislocation movement. As a result of the recovery, the mechanical properties and the structure of the metal observed in the microscopic scale do not change substantially. (Statement in lines 248 - 260 pages 11,12).
However, recrystallization occurs at a higher temperature than
recovery process. The process of dynamic recrystallization is accompanied by thermal activation that accompanies the presence of a single-phase structure of the alloy. In the case of the observed samples, both the device used in the mechanical synthesis process and the results of the phase composition analysis (XRD) do not indicate that the dynamic recrystallization process has started.
These suggestions of Reviewer provide our the new insights and will be establishing new directions of our future research.
- In Tables 1 and 2 as well as in Fig. 4 error bars for the average alloy properties should be added.
Our reply: I thank for this comment. According to the Reviewer’s comments, an error bars was added in Fig.4, Tables 1 and 2. In Table 3 the presentation of the accuracy of the results was standardized.
Thank you again for all your suggestions and comments. I hope you appreciate the specific changes I have made in response to these comments and that overall you feel that the main arguments and contribution are now much stronger.
Sincerely yours,
Sabina Lesz
Reviewer 3 Report
The paper deals with the preliminary study on the effect of the high energy milling times on the quaternary Mg65Zn30Ca4Gd1 alloy, in particular, it aims to define the time of milling time necessary to reach a stable composition homogeneity and microstructure refinement of the alloy.
The results are well presented however the paper could be improved with some minor correction/modifications. The comments are detailed in the revised paper.

Author Response
Dear Reviewer,
Thank you very much for your suggestions. All of them have been taken into consideration in order to improve this paper.
Please note that the referees’ comments are given in italic and my replies are given in bold. The changes in the text of the revised manuscript are highlighted in colour. All my responses to your comments are highlighted using the "Track Changes" function in Microsoft Word.
- “It is not clear the novelty, I suggest to specify specify which properties of the alloy have been investigated” (Page 3 lines: 95, 96).
Our reply: We thank you for this comment. According to the Reviewer’s comments, the text was added. From line 110, page 3:
Hence, the objective of this paper is to present the preliminary study on the effect of the high energy milling times on the quaternary Mg65Zn30Ca4Gd1 alloy. The X-ray diffraction and scanning electron microscopy with energy-dispersive spectroscopy were carried out to study powder morphology and chemical composition of Mg65Zn30Ca4Gd1 powders. The powder size was assessed by granulometric method and the Vickers hardness test was used for microhardness testing.
- Please detail the measurement condition Page 4, line 130-136:
Our reply: The sentence “The phase analysis of the alloy was performed using the PANalytical Empyrean diffractometer with Cu-Ka radiation and PIXCell counter.” has been changed and completed:
The X-ray diffraction analyses were performed with the use of the Empyrean PANalytical Diffractometer (Almelo, The Netherlands) equipped with a copper anode (Ka1 lCu = 1.5418 Å), Ni filter, PIXcell detector. The measurements were made in a step-scanning mode in a range of 10 – 150° 2θ. The phase analysis of substrates and milling products were performed with a High Score Plus PANalytical software (Almelo, The Netherlands) integrated with the ICDD PDF4+ 2016 data base (International Centre for Diffraction Data, Newtown Square, PA USA).
- PLease specify the measuring condition in the material and methods section.
Our reply:
The measurement conditions have been supplemented in the "2. Materials and Methods " section on page 4, lines 130-136 (above).
- Plase report the ICDD reference number adopted for the peaks identification (Page 5, Figure 2).
Our reply: Page 6, figure 2 caption has been completed:
(MgZn2 - 03-065-0120; GdH2 - 01-089-4063; Ca(OH)2 – 00-004-0733; Mg – 00-001-1141; MgO – 00-003-0998; Zn – 03-065-3358)
- Please add the relevant references (Page 5, line 141).
Our reply: We thank for this comment. According to the Reviewer’s comments, the relevant references were added (Page 6, line 170-172).
Amorphization has been observed on a large number of alloy systems below a critical crystallite size of few nanometers having been synthesized by high-energy ball-milling technique [31,32,39].
- Every sample was measured with 5 indentation. (Page 7, line 168). This information should be moved into materials and methods section.
Our reply: According to the Reviewer’s comments, this information was moved to methods section.
- Revise the English language (Page 8, line 172).
Our reply: Thank you very much for your suggestions. The English has been substantially improved (the manuscript has been approved by an English-speaking native). All my responses to your comments are highlighted using the "Track Changes" function in Microsoft Word.
For example:
The sentence:
During this process strong plastic deformation occur due to constant friction welding and fragmentation which results in increase of the hardness value [38].
Was changed to:
During this process strong plastic deformation occurs due to constant friction welding and fragmentation which results in an increase of the hardness value [32,39–43].
- This needs to be argued better (Page 8, line 173):
“Clearly after 30 hours of milling time the hardness is lower that after 20 hours, but it can be correlated to the particle size, as the size after 30 hours is bigger as well”
Appropriate references were added:
Clearly after 30 hours of milling time the hardness is lower that after 20 hours, but it can be correlated to the particle size, as the size after 30 hours is bigger as well [32,39,41].
- Please introduce and describe better the meaning of this figure, the process methods adopted and its stages should be described in materials and methods section:“It can be seen in Figure 4.”( Page 8, line 180)
Our reply: We thank you for this comment. According to the Reviewer’s comments, the figure 4 description was extended. The methods presented in Figure 4, are described in the materials and methods part, as the mentioned figure is just a graphical representation of merged data. The legend has been added.
Was changed to:
It can be seen in Figure 4, which represents the average hardness values as well as particle size for Mg65Zn30Ca4Gd1 powder alloy milled for 13, 20, 30, 40, and 70 hours, respectively.
- Why this should be an expected results? The reader can be not aware or understand the reason reading the paper:“Obviously after 70 hours the particles are smaller than after 13, albeit the hardness is comparable”.( Page 8, line 185)
Our reply: It was changed into:
After 70 hours the particles are smaller than after 13, albeit the hardness is comparable.
- For a correct comparison the figure should be reported at the same magnification (Figure 5).
Our reply: The Figures were changed accordingly with your comment. They are of the same magnification now.
Thank you again for all your suggestions and comments. I hope you appreciate the specific changes I have made in response to these comments and that overall you feel that the main arguments and contribution are now much stronger.
Sincerely yours,
Sabina Lesz

Reviewer 4 Report
Ref.comments to the paper titled as “Characteristic of the Mg-Zn-Ca-Gd alloy after mechanical alloying” written by Sabina Lesz, BartÅ‚omiej Hrapkowicz, MaÅ‚gorzata Karolus, Klaudiusz GoÅ‚ombek
It is known that currently a special role is assigned to materials that can be used for biomedicine, for example, as implants. A lot of qualities, including the reactive ability to interact with different elements, should be taken into account when choosing such materials. In this regard, magnesium-based materials have their own advantages. From this point of view, the paper is actual and modern.
It is very good that the authors have presented nice literature analysis of the problem and have indicated their own place among other investigations. Useful schematic representation of the experimental process has been shown in the session “Materials and methods”.
X-ray diffraction analysis, particle size histograms, average particle size for samples obtained for the different milled time, estimated hardness value for samples obtained for the different milled time, illustration of the average hardness values as well as particle size for Mg-based structure studied alloy milled for different hours have been shown.
Important data about morphology with the micrographs of the Mg65Zn30Ca4Gd1 alloy powder milled for different hours can extend our knowledge about this structure.
The conclusion corresponds to the proposed text and contains its main conclusions. The list of articles includes recent works in the direction of studying similar systems.
I for my local opinion, I can recommend the paper to be published in the present form!

Author Response
Extensive editing of English language and style required.
Our reply: Thank you very much for your suggestions. The English has been substantially improved (the manuscript has been approved by an English-speaking native). All my responses to your comments are highlighted using the "Track Changes" function in Microsoft Word.
For example in lines: 78 (page 2), 114, 116, 117 (page 3), 134, 135, 144, 146, 150 (page 4), 152 (page 5). The rest English correct was mentioned in the text in color.
Thank you again for all your comments. I hope you appreciate the specific changes. I have made in response to these comments and that overall you feel that the main arguments and contribution are now much stronger.
Round 2
Reviewer 2 Report
The authors satisfactorily addressed my comments. I can now recommend publication of the paper.
Author Response
Thanks for the review.
Sincerely yours,
Sabina Lesz